# Biodistribution of Nanostructured Lipid Carriers in Mice Atherosclerotic Model

**DOI:** 10.3390/molecules24193499

**Published:** 2019-09-26

**Authors:** Laurent Devel, Gunter Almer, Claudia Cabella, Fabrice Beau, Mylène Bernes, Paolo Oliva, Fabrice Navarro, Ruth Prassl, Harald Mangge, Isabelle Texier

**Affiliations:** 1CEA, Institut des Sciences du Vivant Frédéric Joliot, Service d’Ingénierie Moléculaire des Protéines (SIMOPRO), Université Paris-Saclay, 91190 Gif-sur-Yvette, France; laurent.devel@cea.fr; 2Clinical Institute of Medical and Chemical Laboratory Diagnostics, Medical University of Graz, 8036 Graz, Austria; gunter.almer@medunigraz.at; 3Centro Ricerche Bracco, Bracco Imaging SpA, 10010 Colleretto Giacosa, Italy; claudia.cabella@bracco.com; 4University Grenoble Alpes, CEA, LETI, F-38000 Grenoble, France; 5Institute of Biophysics, Medical University of Graz, 8036 Graz, Austria; ruth.prassl@medunigraz.at

**Keywords:** nanomedicine, lipid nanoparticles, atherosclerosis, ApoE^-/-^ model, macrophage elastase (MMP-12), active targeting

## Abstract

Atherosclerosis is a major cardiovascular disease worldwide, that could benefit from innovative nanomedicine imaging tools and treatments. In this perspective, we here studied, by fluorescence imaging in ApoE^-/-^ mice, the biodistribution of non-functionalized and RXP470.1-targeted nanostructured lipid carriers (NLC) loaded with DiD dye. RXP470.1 specifically binds to MMP12, a metalloprotease that is over-expressed by macrophages residing in atherosclerotic plaques. Physico-chemical characterizations showed that RXP-NLC (about 105 RXP470.1 moieties/particle) displayed similar features as non-functionalized NLC in terms of particle diameter (about 60-65 nm), surface charge (about −5 — −10 mV), and colloidal stability. In vitro inhibition assays demonstrated that RXP-NLC conserved a selectivity and affinity profile, which favored MMP-12. In vivo data indicated that NLC and RXP-NLC presented prolonged blood circulation and accumulation in atherosclerotic lesions in a few hours. Twenty-four hours after injection, particle uptake in atherosclerotic plaques of the brachiocephalic artery was similar for both nanoparticles, as assessed by ex vivo imaging. This suggests that the RXP470.1 coating did not significantly induce an active targeting of the nanoparticles within the plaques. Overall, NLCs appeared to be very promising nanovectors to efficiently and specifically deliver imaging agents or drugs in atherosclerotic lesions, opening avenues for new nanomedicine strategies for cardiovascular diseases.

## 1. Introduction

Lipid-core nanocarriers (typically 50–300 nm diameter) have been developed since the 1990s, initiated by the groups of Müller [1,2] and Gasco [3]. They are colloidal dispersions comprising of lipid core nanodroplets stabilized in aqueous buffer by a single layer of surfactants (e.g., phospholipids, poly(ethylene glycol)-based (PEGylated) surfactants, amphiphilic saccharides). From originally pure solid lipid core nanoparticles, the addition of liquid lipids to design nanostructured lipid carriers has improved drug payload and colloidal stability of the objects, with better control over the release of active ingredients [4]. Nanostructured lipid carriers (NLC) have since then been increasingly used in advanced innovative drug formulations [5,6,7,8]. NLC ingredients are non-toxic and highly biocompatible, approved by the FDA (Food and Drug Administration) or the EMA (European Medicines Agency) [7], and NLC safe profile has been underlined in many studies [9]. NLC can be produced by different processes, such as micro emulsification, solvent emulsification/evaporation, ultra-sonication, or high-pressure homogenization (HPH). HPH is the preferred solvent-free method to translate production from the lab scale to an industrial process [5,7]. Interestingly, lipid nanoparticles including NLC can be characterized by a set of physicochemical techniques providing morphological, structural, core crystallinity, colloidal stability, and drug payload information [10], which can be used to tailor drug release kinetics [11].

NLC can be administered through various delivery routes to serve different therapeutic purposes, and are particularly relevant for oral [12], ocular [13], dermal [14], and parenteral [15,16] delivery. Lipid nanoparticles with different compositions and size have been proposed for the parenteral delivery of chemotherapeutic drugs (e.g., 5-fluoroacil, doxorubicine, paclitaxel) for cancer therapy [17], for brain targeting and the treatment of central nervous system diseases (e.g., clozapine, quercetin, bromocriptine) [16,18], for vaccination [19], or for gene therapy [20]. However, their use has been scarcely described for vascular diseases, a new frontier for nanomedicine [21,22]. Colloidally stable solid lipid nanoparticles comprised of nucleolipids were loaded with iron oxide particles and PG12 prostacyclin, and reduced in vitro platelet aggregation [23]. Lipid nanoparticles prepared by nanoprecipitation/solvent diffusion method and loaded with lovastatin were shown to accumulate in macrophages and foam cells present in atherosclerotic plaques [24]. Maranhão’s group developed paclitaxel-oleate [25] and carmustine [26] loaded lipid nanoparticles produced by a microfluidization process, and showed that their intravenous administration could reduce atherosclerosis lesions in rabbits. Slightly different nanoparticles, based on a perfluorocarbon oily core, were developed in Wickline’s group for MRI atherosclerosis imaging [27] but also for therapeutic purposes [28].

In parallel, biodistribution experiments in rodents and dogs demonstrated that NLC developed in our group, comprised of a core of soybean oil and Suppocire™NB, and a shell of lecithin and Myrj™s40 (a PEGylated surfactant with 40 ethylene(glycol) units), were metabolized by the hepatobiliary pathway, and significantly accumulated in lipid-processing areas, such as those implicated in steroid hormone synthesis (follicles and corpus luteum of ovaries, X zone of adrenals) [29,30]. This peculiar pattern could be due to the nanometric size and lipid nature of the particles, conferring them some structural similarity with blood-circulating very-low-density lipoproteins (VLDL) and chylomicrons. Until now, other recombinant or engineered lipoprotein classes, LDL (low-density lipoproteins) [31] and HDL (high-density lipoproteins) [32,33,34,35,36,37], have been proposed as imaging and drug delivery platforms for atherosclerosis. NLC could therefore present high interest for cardiovascular applications as synthetic mimics of lipoproteins.

Due to the multiplicity of biological processes involved in the development of atherosclerotic plaques, different targeting approaches have been proposed in the literature to specifically increase the uptake of nanovectors in atherosclerotic lesions through different nanoparticle surface functionalization [38,39]. Nanoparticles could be directed towards plaque cellular targets, such as endothelial cells, monocytes, macrophages, and platelets, and plaque extra-cellular components, such as collagen, fibrin, or lipids. Endothelial cells expressing inflammatory molecules (E-selectin, P-selectin, ICAM-1, VCAM-1, etc.), have been the main targets for a large range of nanoparticles functionalized with antibodies, sialyl Lewis derivatives, or peptide sequences [38]. Scavenger receptors of macrophages (especially MSR-1, SR-B1, or CD-36) have also been targeted by apolipoprotein A, phosphatidylcholine, triphenylphosphonium, or dextran decorated nanoparticles [38,40].

In the present study, the in vivo biodistribution of NLC in an ApoE^-/-^ atherosclerotic mice model was investigated through fluorescence imaging. The dye (DiD)-loaded nanoparticles were either plain or surface-functionalized by a highly potent and selective inhibitor of MMP-12 named RXP470.1 [41]. MMP-12, also named macrophage elastase, is a metalloprotease known for its ability to cleave elastin, a major component of elastic laminae in the media of arteries. This enzyme is produced by macrophages that mainly contribute to inflammatory processes. As such, MMP-12 is upregulated in several inflammatory diseases, including abdominal aortic aneurysm [42] and atherosclerosis [43]. In the ApoE^-/-^ atherosclerotic mice model, it has been demonstrated that MMP-12-secreting macrophages supported atherosclerotic plaque expansion [44]. Importantly, a pseudo peptide, RXP470.1, has been able to selectively target MMP12 and demonstrated its capacity to selectively block MMP12 proteolytic activity in several preclinical models [45,46,47,48,49,50]. In an ApoE^-/-^ mice model of atherosclerosis, this compound induced a change in atherosclerotic plaques composition, thus limiting their evolution towards an instable phenotype, strongly suggesting MMP12 as a relevant target in this model. In addition, optical imaging agents with a structure derived from RXP470.1 showed excellent in vivo performances of targeting MMP12 within carotid arteries in a mouse model of aneurysm [51]. In this context, developing NLC exposing RXP470.1 on their surface (RXP-NLC) may constitute a promising approach to achieve simultaneously selective targeting of MMP12-secreting macrophages for subsequent drug delivery, and blockage of MMP-12 activity.

## 2. Results

### 2.1. Synthesis and Characterization of Nanostructured Lipid Carriers

To obtain RXP-NLC, RXP470.1 pseudo-peptide containing a maleimide group (RXP470.1-PEG_2_-NH_2_-Mal (**4**), see Appendix A for detailed synthesis) was coupled through thioether linkage to DiD-loaded NLC previously functionalized with thiol-protected groups (NLC-S-S-Pyr) (Figure 1). Purified NLC-S-S-Pyr were obtained by the formulation of NLC according to previously described protocols [52,53] in the presence of additional functional SA-PEG_100_-S-S-Pyr surfactant, then dialysis. The SA-PEG_100_-S-S-Pyr surfactant (**1**), of which synthesis is described in Appendix A, comprised a stearic moiety (SA) for anchoring in the lipid core, a thiol-protected function (-S-S-Pyr: 2-pyridil-dithio) for further NLC functionalization, and a PEG chain of 100 ethylene(glycol) units for presentation of the functional group “above” the Myrj™ s40 PEG layer. The deprotection of the 2-pyridil-dithio functions was performed in the presence of DTT (dithiothreitol) (Figure 1a) before grafting of RXP470.1-maleimide pseudo-peptide (Figure 1b) to yield the RXP470-modified NLC. After RXP470.1 grafting onto the nanoparticle surface, the remaining free thiol groups were neutralized by conversion to hydroxyl groups by grafting of 1-(2-hydroxyethyl)-1H-pyrrole-2,5-dione to obtain RXP-NLC (Figure 1c). Control nanoparticles without RXP (NLC) were obtained by the direct neutralization of the thiol groups of NLC-SH by 1-(2-hydroxyethyl)-1H-pyrrole-2,5-dione (Figure 1d). At the end of the process, both NLC and RXP-NLC particles were purified by dialysis using 12,000–14,000 g/mol MW cut-off membranes. Alternatively, size exclusion chromatography (SEC) performed on Sephadex G50 gel columns eluted with PBS (10 mM phosphate, pH 7.4) could be used.

Hydrodynamic diameter and zeta potential of NLC-S-S-Pyr, NLC, and RXP-NLC nanoparticles were characterized by dynamic light scattering (DLS) (Figure 2a,b). All dispersions displayed nanoparticles of about 60–65 nm hydrodynamic diameter, and a polydispersity index below 0.2. Particle size was confirmed by SEM (Figure 2c,d). NLC and RXP-NLC particles were colloidally stable for more than three months when stored in 1X PBS at a concentration of 100 mg/mL (Appendix A). This long-term stability was ensured by the PEG coating which provided a steric barrier preventing particle coalescence [53], and a particle surface close to neutrality (zeta potential absolute value < 15 mV).

The number of DiD per particle was calculated from the visible absorption measurement at 650 nm [52]. The number of ligands per RXP-NLC was estimated from the concentration of both RXP470.1 pseudo peptide and lipids in solution. The RXP470.1 concentration was determined by measuring the RXP-NLC composition in glutamate residues through amino acid dosing, while the lipid concentration was quantified by weighting lyophilized particle samples of known volume. Taken together, these data suggest a mean number of ligands of 105 ± 15 RXP470.1/particle, corresponding to a density of about 1 ligand/ 105–140 nm^2^. A maleimide/thiol coupling yield of 60% could thus be deduced. This direct quantitation of RXP470.1 bound to the surface of the NLC was in the same range as the indirect less precise quantitation obtained by the HPLC measurement of free unbound RXP470.1 eluted from SEC column during RXP-NLC purification (giving 49% yield for RXP grafting onto NLC). All the particle characterization results are summarized in Table 1.

### 2.2. In Vitro Characterization of Nanostructured Lipid Carriers

The binding capacity of the RXP470.1 ligand grafted on RXP-NLC surface was first assessed through an inhibition assay in the presence of a fluorogenic substrate [54]. The affinity and selectivity profile of RXP-NLC was determined on a set of eight human MMPs and compared to the parent ligand. Although RXP-NLC displayed a Ki value 10-fold lower than that of the parent RXP470.1, its selectivity profile remained largely favorable for hMMP12 (Table 2).

The colloidal stability of DiD-loaded NLC and RXP-NLC in human serum was evaluated by DLS and fluorescence measurements performed at different times after particle incubation at 37 °C at a concentration of 5 mg of particles/mL of serum (Figure 3). Colloidal stability in PBS was also measured as control. The particle hydrodynamic diameter remained constant for 24 h when diluted particles were incubated at 37 °C in PBS buffer, but about 50% DiD fluorescence was lost in 24 h due to partial dye leakage or degradation in such diluted medium at 37 °C. This was consistent with previous data [55]. NLC and RXP470-NLC were stable for two hours at 37 °C in human serum, since DLS particle profiles were not significantly modified during this period of time. At 3 h after particle incubation, particle DLS peaks were enlarged and Z-average shifted to higher values (Figure 3a). After 4 h, no peak could confidently be attributed to the nanoparticles and only large background noise due to serum proteins was observed. This corresponded with increased fluorescence loss of the nanoparticles, which was indicative of their destabilization (Figure 3b). Overall, these results indicated that both NLC and RXP470-NLC were stable in serum for at least two hours.

### 2.3. Pharmacokinetics and Biodistribution of Nanostructured Lipid Carriers in ApoE^-/-^ Mice

The in vivo fate of unmodified lipid nanoparticles (NLC) and RXP470.1-functionalized ones (RXP-NLC) was investigated in ApoE^-/-^ mice fed with a high-fat rodent diet for 10 weeks to develop mature atherosclerotic plaques. The particle blood clearance and biodistribution were evaluated following intravenous administration of either NLC or RXP-NLC at a dose of 5 mg/mouse. For each set of nanoparticles, pharmacokinetics was established through the collection of blood samples at different time points and quantification was performed through fluorescence measurement at appropriate DiD excitation and emission wavelengths. As shown in Figure 4a, NLC and RXP-NLC displayed very similar blood clearance profiles, with a blood plasma half-life of 1 h for both particles. This rapid distribution phase was followed by a slower decline of the plasma concentration, which led to a non-negligible portion of fluorescence remaining 24 h after injection (10%). At this time point, organs (liver, spleen, kidneys, lungs, femur bone, and brain) were harvested and tissue fluorescence was evaluated. As illustrated in Figure 4b, particles mainly accumulated in liver, which was a hallmark of mainly hepatic-biliary metabolism, while fluorescence in other organs was quite low. This biodistribution pattern was not significantly affected by RXP470.1 functionalization, and was consistent with previously published data for this type of lipid nanoparticles [29,55].

For both types of particles, the ex vivo the isolated aorta harvested 24 h after animal sacrifice and exsanguination were imaged (Figure 5a). The 2D fluorescence imaging at appropriate DiD excitation and emission wavelengths allowed to visualize a detectable signal within atherosclerotic plaques independently of nanoparticle type. Imaging of the whole aorta indicated a mean uptake corresponding to 0.21 ± 0.06% of the injected dose. In addition, fluorescence counting of a region of interest (ROI = 1 mm^2^) drawn on the brachiocephalic artery confirmed a comparable accumulation between non-functionalized NLC and RXP470.1-NLC (Figure 5b). In parallel, a histological analysis of brachiocephalic artery cross-sections, dissected 48 h after particle injection, was also performed by confocal fluorescence microscopy. As shown in Figure 5c, a significant NLC uptake in the brachiocephalic artery plaques was observed for both non-targeted and RXP-targeted nanoparticles in the macrophage-containing area on the surface of the plaques and the fibrous cap below. Such staining patterns were not observed in control mice injected with PBS (Appendix A). Histology for a qualitative representation of nanoparticle biodistribution also confirmed strong NLC accumulation in the liver, as well as in the lymph node located next to the brachiocephalic artery (Appendix A).

## 3. Discussion

As underlined in the introduction, the potentialities of lipid core nanoparticles such as nanostructured lipid carriers to target atherosclerotic lesions have remained largely unexplored, despite few promising results [24,25,26,27,28]. However, NLC display suitable features in terms of particle size, composition, in vivo tolerance, and potential translation to the clinic (easily up-scaled production at a very reasonable cost) for nanomedicine application in the cardiovascular field. In this context, NLC and RXP470.1-NLC were synthetized here, and we thoroughly characterized both nanoparticles and their biodistribution pattern in the ApoE^-/-^ atherosclerosis mice model.

Both nanoparticles presented similar physico-chemical characteristics, with particle size (60–65 nm diameter), surface charge (–10 to –5 mV zeta potential), and PEGylated surface favorable to a long blood circulating time [56]. Indeed, NLC and RXP-NLC displayed 2 h stability in human serum in vitro. After intravenous injection in ApoE^-/-^ mice, a rapid biodistribution phase with blood half-life of approximately 1 h, followed by slower decline resulting in about 10% of particles still in circulation 24 h after intravenous injection was observed, which was in line with previously published data on similar lipid nanoparticles [29]. NLC mainly accumulated in the liver, whereas fluorescence signal in other major organs (spleen, kidneys, lungs, brain) remained very low. Hepatobiliary metabolism is thought to be the main clearance pathway for NLC [29]. Nanoparticles also accumulated significantly in atherosclerotic plaques 24 h after injection. Histological images after 48 h of injection indicated the localization of the NLC mainly on the lumen side of the plaques, where active macrophages take up the particles, with some further internalization up to approximately 200 µm deep into the plaque, which represented the area of the fibrous cap. These observations tend to show that this type of nanoparticles have affinity, after blood injection, towards body lipid-processing areas, as was suggested in a previous biodistribution study performed in healthy and cancer model mice [29]. The observed uptake of NLC and RXP-NLC in atherosclerotic lesions promotes the interest of such nanocarriers for therapeutic approaches in cardiovascular diseases.

In this study, we also explored an active targeting strategy for NLC relying on particle functionalization with a RXP470.1 motif. RXP470.1 has been developed as a highly potent and selective inhibitor of MMP12, an extra-cellular matrix target produced by inflamed macrophages [41]. The maleimide function was added to the C-terminal end of the pseudo peptide since it has been previously demonstrated that comparable modification did not impair the RXP470.1 binding properties [51,54]. A PEG spacer was also incorporated to maintain the targeting ligand accessibility for interaction with its privileged target. The grafting of the RXP470.1 peptide on the NLC surface was carefully controlled and quantified (105 ± 15 ligands/particle), both by complementary direct and indirect methods, and did not significantly impact the physico-chemical properties of the NLC. In vitro, RXP-NLC displayed a 10-fold lower affinity than the parent molecule for MMP-12, but its selectivity profile remained largely favorable for this enzyme. Surprisingly, RXP-NLC even displayed better selectivity factors than the parent RXP470 towards hMMP-2, 8, 9, and 13. This affinity/selectivity profile was comparable to that of previously reported RXP470-derived probe [51,54], thus validating our particle design. Interestingly, the presence of RXP470 did not significantly modify the pharmacokinetics and biodistribution pattern of NLC, resulting in a comparable accumulation between RXP-NLC and NLC within atherosclerotic plaques. This undoubtedly constitutes an encouraging and valuable result. Indeed, the presence of RXP470 ligand could have increased the NLC opsonization with detrimental effect both on blood clearance profile and uptake within targeted tissues, as observed with other targeting antigens [57,58].

However, the targeting capability of RXP-NLC did not appear superior to that of uncoated NLC. Several factors including RXP-NLC affinity towards its privileged target, particle size, and ligand accessibility, could explain this disappointing result. In regards to the affinity issue, further optimization of PEG spacer length and ligand surface density could be implemented to restore a sub-nanomolar affinity of RXP-NLC towards MMP-12. However, a gain in affinity in vitro would not ensure an improved uptake within the targeted tissues in vivo, and the precise affinity at which maximum tissue uptake can be achieved remains difficult to predict. In vivo, the targeting capacity of an antigen is often related to its size and the nature of targeted tissues. In case of small targeting probes, unbounded molecules are rapidly cleared from the tissues following their initial uptake, while bounded molecules are retained. Accordingly, a high level of specificity can be achieved when high affinity molecules are used. In contrast, several tumor targeting studies have shown that macromolecules in the size range of nanoparticles (radius >20 nm) could display similar tumor levels whether targeted or not. In this case, the uptake is mainly governed by passive enhanced permeability and retention (EPR) effect, with a slow clearance rate of unbounded particles [59]. A similar phenomenon could take place when NLC accumulate within atherosclerotic plaques, which could hinder the visualization of the targeting RXP moiety. In addition, we have previously demonstrated that RXP470 ligand tends to bind serum albumin [54]. This may also occur with RXP-NLC, leading to a fraction of targeting motifs potentially inaccessible for binding to MMP12, thus impacting the targeting capacity of the particles.

On the other hand, the density of MMP12 targets may also be a limiting factor. Indeed, uncontrolled matrix metalloprotease (MMP) activity could have disastrous effects on the microenvironment, which is why MMPs activation is tightly controlled in vivo. Thus, over-expression of MMPs reported in numerous pathological tissues mainly refers to MMPs under their inactive zymogen and inhibitor-bound forms and only a small fraction of active form seems to be actually present [60]. Thus, we do not exclude that the amount of active MMP12 within atherosclerotic plaques, the sole form targeted by RXP470 ligand, may not be sufficient to induce a significant uptake of RXP-NLC relative to that of NLC.

Overall, particle accumulation seemed to be mainly driven by a passive EPR effect or a lipid metabolism mechanism, similar to what is observed for HDL or LDL lipoproteins [39]. However, the accumulation of NLC displaying RXP470 on their surface may allow inhibition of MMP12 activity, which could contribute to atherosclerotic plaque stabilization. This hypothesis is reinforced by the observation of RXP-NLC and uncoated NLC uptake in the plaque area rich in macrophages that constitute the main source of MMP12 [61]. In this respect, a strategy that would consist of developing bifunctional NLC able to both achieve anti-inflammatory drug delivery and MMP12 blockade remains valuable to limit plaque weakening. This approach is currently under investigation, but these preliminary results already outlined the interest of NLC for atherosclerotic plaque therapy.

## 4. Materials and Methods

### 4.1. Materials

Suppocire NB™ was purchased from Gattefossé (Saint-Priest, France), Lipoid™ S75 (soybean lecithin at >75% phosphatidylcholine) from Lipoid (Ludwigshafen, Germany), Myrj™ S40 (polyethylene glycol 40 stearate) and super-refined soybean oil from Croda Uniqema (Chocques, France), and DiD (1,1′-dioctadecyl-3,3,3′,3′-tetramethylindodicarbocyanine perchlorate) from Fisher Scientific (Les Ulis, France). The synthesis of SA-PEG_100_-S-S-Pyr and RXP470.1-maleimide (a RXP470.1 derivative with a PEG extension and maleimide moiety in its C-terminal end) are described in Appendix A. Other chemicals were purchased from Sigma-Aldrich (Saint-Quentin-Fallavier, France). Nanoparticles were filtered through sterile Acrodisc^®^ syringe filters (0.2 µm, 13 mm) with Supor^®^ membrane (PALL Corporation, USA) for sterilization.

### 4.2. Synthesis of Nanostructured Lipid Carriers

Nanostructured lipid carriers were prepared similarly to previously described protocols, by mixing prepared lipid and aqueous phases at 50 °C prior to ultrasonication [24,31]. The lipid phase comprised of 255 mg of soybean oil, 85 mg of Suppocire™ NB, 65 mg of lecithin, and 800 nmol (770 µg) of DiD, and the aqueous phase was composed of 335 mg of Myrj™ S40, 10 mg of SA-PEG_100_-S-S-Pyr (2 µmol), and 1X PBS buffer (qsp 2 mL). After preparation and heating of the lipid and aqueous phases at 50 °C, the two phases were mixed and sonicated for 5 min as previously described (VCX750 Ultrasonic processor, Sonics, Newtown, USA) [24,31]. The nanoparticle dispersions were then purified by dialysis against 1X PBS (10 mM phosphate, pH 7.4) using 12−14 kDa molecular weight cutoff membranes (ZelluTrans, Roth, France). Particle concentration was measured at the end of the dialysis step by weighting lyophilized samples of known volume (with buffer salt weight taken into account). The NLC-S-S-Pyr dispersion was then treated according to conditions described in Figure 1. DTT (4 mg, 25.9 µmol, for a batch of 750 mg of particles) was added for deprotection of the thio-pyridil group (Figure 1a). After 2 h incubation at room temperature, NLC-SH were dialyzed overnight (12,000–14,000 MWCO, PBS buffer) to eliminate excess DTT. For RXP-NLC synthesis, 720 nmol of RXP470.1-maleimide in 180 µL PBS were then incubated with 300 mg of particles (approximately in 3 mL of buffer), and the mixture was stirred for 4 h at room temperature, then overnight at 4 °C (Figure 1b). Then, 1 µmol (140 µg prepared in 40 µL PBS) of 1-(2-hydroxyethyl)-1H-pyrrole-2,5-dione (for 300 mg of particles) was added and the mixture was incubated for 30 min at room temperature to neutralize by a non-reactive hydroxyl coating non-reacted thiol functions on particle surface (Figure 1c). For control NLC synthesis, 2 µmol (280 µg prepared in 80 µL PBS) of 1-(2-hydroxyethyl)-1H-pyrrole-2,5-dione (for 300 mg of particles) were added to the dispersion of NLC-SH and the mixture incubated for 30 min at room temperature to neutralize by a non-reactive hydroxyl coating thiol functions on the particle surface (Figure 1d). RXP-NLC and NLC were finally purified by dialysis for 24 h (12,000–14,000 MWCO, PBS buffer). RXP-NLC could also be purified by SEC on Sephadex 50 gel phase using 1X PBS for elution. DiD-labelled particles were eluted in the 1–4.5 mL fractions, as assessed by fluorescence quantification (Tecan fluorescence microplate reader), whereas unbound RXP470 were eluted in 7–17 mL fractions (HPLC quantification) (Appendix A). The purified NLC and RXP-NLC dispersions were diluted to a concentration of 100 mg/mL and filtered through a 0.22 μm Millipore membrane for sterilization before storage and use.

### 4.3. Lipid Nanoparticle Characterizations

Dynamic light scattering (DLS) was used to determine the particle hydrodynamic diameter and zeta potential (Zeta Sizer Nano ZS, Malvern Instrument, Orsay, France). Particle dispersions were diluted to 2 mg/mL of lipids in 0.22 µm filtered 0.1 X PBS and transferred in Zeta Sizer Nano cells (Malvern Instrument) before each measurement, performed in triplicate. Results (Z-average diameter, polydispersity index, zeta potential) were expressed as mean and standard deviation of three independent measurements performed at 25 °C.

TEM images of NLC were obtained after negative staining (2% uranyl acetate) similarly to a previously published protocol [62].

Particle concentration (expressed in mg of lipids/mL) was determined by weighting lyophilized dispersion samples of known volumes (taking into account the weight of buffer salts). Absorbance measurements (Cary UV-visible spectrophotometer, Les Ulis, France) at 650 nm performed on particle dispersions diluted to 10 mg/mL of lipids allowed the quantification of the number of DiD dyes encapsulated per NLC (encapsulation yield of DiD > 95%) [52]. The number of RXP470.1 pseudo-peptides grafted on the particle surface was quantified by dosing the RXP-NLC composition in glutamate residues. Briefly, an RXP-NLC sample was vacuum dried, sealed in a glass tube using a PicoTag system (Waters Associates, Milford, MA), and hydrolyzed under vapor phase of 6 N HCl with a crystal of phenol for 17 h at 110 °C. The hydrolyzed sample was then dissolved in 20–50 µL of MilliQ water and 5–20 µL of the HCl hydrolysate was subsequently analyzed and quantified via ninhydrin derivatization on an aminoTac JLC-500/V amino acids analyzer (JEOL, Japan). A calibration with an amino acids standard H solution was performed at the beginning of analysis. In addition, HPLC quantification of free unbound RXP470.1 eluted from SEC column during RXP-NLC purification (7–17 mL fractions) was also performed (see Appendix A) and confirmed results of direct RXP470.1 quantitation.

### 4.4. Enzymatic Activity

Affinity and selectivity profile of RXP-NLC were determined toward a set of 8 human MMPs. Human MMP-8, -9, -12, and -13 were produced at CEA in Saclay according to a procedure previously reported [63]. Other MMPs were purchased from R&D Systems (Minneapolis, MN, USA). MMP inhibition assays were carried out in 50 mM Tris-HCl buffer, pH = 6.8, 10 mM CaCl_2_ at 25 °C as previously described [63]. Pro-MMPs were pre-activated by p-aminophenylmercuric acetate following the method described by R&D Systems. Titration experiments were carried out to determine the active enzyme concentration for each MMP prior to the assay [41]. Continuous kinetic assays were performed by recording the fluorescence increase induced by the cleavage of fluorogenic substrates (Mca-Pro-Leu- Gly-Leu-Dpa-Ala-Arg-NH_2_ for MMP-1, -2, -7, -8, -9, -10, -12, -13, and -14 and Mca-Arg-Pro-Lys-Pro- Val-Glu- Nva-Trp-Arg-LysDNP-NH_2_ for MMP-3). A typical experiment was performed in 200 μL of Tris buffer containing 0.2–0.5 nM of MMP, and a 4.5 μM concentration of fluorogenic substrates. Black, flat-bottomed, 96-well non-binding surface plates (Corning-Costar, Schiphol-RijK, Netherlands) were used for these tests. Fluorescence signals were monitored using a Fluoroskan Ascent photon counter spectrophotometer (Thermo-Labsystems, Courtaboeuf, France) equipped with a temperature control device and a plate shaker. For each evaluated compound, the inhibition percentage was determined at five concentrations in triplicate, within the range between 20–80. Ki values were determined using the method proposed by Horovitz and Leviski [64].

### 4.5. Stability in Human Serum

Blood samples were collected from five consenting donors and incubated with EDTA (provided by the Etablissement Français du Sang, CHU, Grenoble, France). Sera were recovered as the supernatant of 2 consecutive centrifugation steps performed at 3000 g (10 min) then 2000 g (15 min), pooled, and used immediately. Then, 25 μL of NLCs (100 mg/mL of lipids) were mixed with 475 µL of serum (final 5 mg/mL of lipids) and put into a shaking water bath with a temperature control set at 37 °C. Samples (75 μL) were withdrawn at 1 h, 2 h, 3 h, 4 h, 24 h, diluted in water (1425 μL), and immediately analyzed by DLS.

### 4.6. In Vivo Imaging Experiments in Mice

Pharmacokinetics and bio-distribution studies were performed in compliance with the National Animal Welfare Regulations at CEA in Saclay. The experiments were approved by the local ethics committee for animal experimentation (Authorization number: 15-056, expiration date: 05/12/2020). ApoE^-/-^ mice (Charles River Laboratories, L’Abrésie, France) were individually housed in polycarbonate cages in a conventional animal facility and had access ad libitum to food and drink. ApoE-knockout mice C57BL/6 (8 weeks old) were fed with a high-fat rodent diet for 10 weeks to develop mature atherosclerotic plaques in the brachiocephalic artery. Then, 5 mg of nanoparticles (NLC or RXP-NLC) were injected intravenously in the tail vein (*n* = 3 for each group) under anesthesia (2% isoflurane). At various time points (0, 1, 5, 15, 60, 90, 210, and 1440 min), blood samples (whole blood) were collected from the retro orbital sinus under anesthesia with 2% isoflurane. Animals were sacrificed by carbon dioxide asphyxiation after 24 h and the organs (liver, kidneys, spleen, lungs, femur bone, and brain) were harvested post-mortem. Blood, organs, and whole aorta were imaged by fluorescence reflectance using FMT 1500′s planar imaging capability (Perkin Elmer, Waltham, MA). Imaging was conducted with the appropriate excitation (Ex) and emission (Em) filter sets: Ex/Em = 675/720 nm. Blood fluorescence was converted to % injected dose per gram (%ID/g) using a standard curve derived from serial dilutions of each probe in mouse blood. Tissue fluorescence was quantified as mean efficiency per pixel and was presented as an arbitrary unit (AU).

### 4.7. Histology

Animal experiments were approved by the Ministry of Science and Research, Austria. Sixteen-week-old female ApoE-deficient mice with a C57Bl/6J genetic background (Charles River Laboratories, Brussels, Belgium) were fed with a Western-type 21% XL (raw fat) experimental food (Ssniff Spezialdiaeten GmbH, Soest, Germany) for 8 weeks. Then, the mice received an intravenous (iv) injection of 5 mg NLC or RXP-NLC/mL blood volume into the orbital vein except negative control. For each mouse, 48 h post-injection, the aortic arch with the brachiocephalic aorta was dissected, washed with PBS (+0.9% NaCl, pH 7.4), blocked with 1% BSA in PBS for 30 min, and subsequently incubated with 5 μg/mL of an AlexaFluor (AF) 488 pre-labeled rat anti-mouse antibody directed against CD68 (AbDSerotec, Dusseldorf, Germany) except negative control. Hoechst 33342 fluorescence dye (1 μg/mL, Invitrogen, Vienna, Austria) was added to every specimen for 15 min to stain the cell nuclei. Then, the specimens were washed with PBS twice, embedded in Tissue Tek O.C.T T COMPOUND (VWR International, Vienna, Austria), and frozen at −20 °C in a cryotom (Microm, Walldorf, Germany). Slices of 10 μm thickness were cut, transferred to SuperFrost^®^ Plus glass slides (Thermo Scientific, Germany), and visualized immediately using an Olympus BX51 Basic Fluorescence Microscope (Hamburg, Germany), operating with a DP71 camera. All images were collected using an Uplanfl 20 × objective. For the visualization of the nanoparticles, a red fluorescence filter with 590–650 nm excitation and 662–738 nm emission range was used. A MWIBA2 filter with 460–490 nm excitation and 510–550 nm emission range was used to visualize the anti-CD68 Ab and auto-fluorescence at the same position of the samples (green fluorescence). A blue fluorescence filter with 360–370 nm excitation and 420–460 nm emission range was used for the visualization of the Hoechst nucleus staining. Overlapping fluorescence images were generated with the Olympus cell^D software.

### 4.8. Statistical Analysis

Data are presented as mean ± standard error (SE). Significance was set at the 0.05 level.

## Figures and Tables

**Figure 1 molecules-24-03499-f001:**
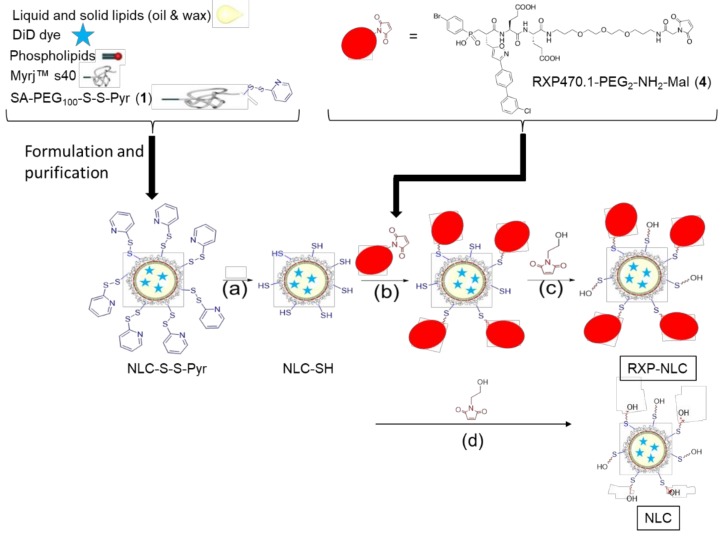
Synthesis of RXP470-modified nanostructured lipid carriers (NLC) and control NLC. (**a**) Addition of dithiothreitol (DTT), incubation 2 h at room temperature, then dialysis overnight. (**b**) Addition of RXP470.1-maleimide, 4 h incubation at room temperature, then overnight at 4 °C. (**c,d**) Addition of 1-(2-hydroxyethyl)-1H-pyrrole-2,5-dione, 30 min incubation at room temperature, then dialysis for 24 h.

**Figure 2 molecules-24-03499-f002:**
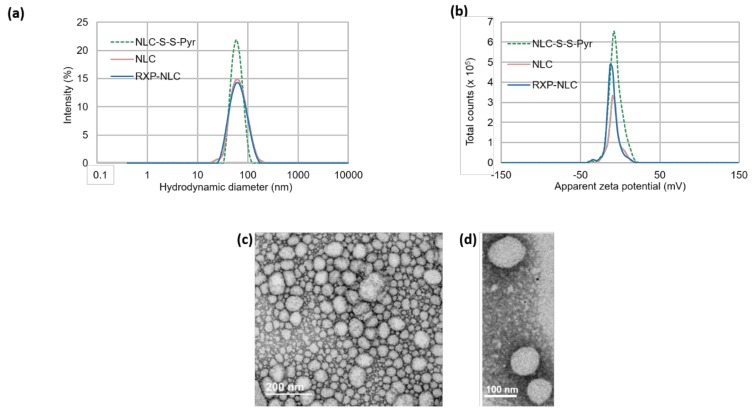
Dynamic light scattering (DLS) and SEM characterizations of nanoparticles. (**a**) Size distribution by intensity of NLC-S-S-Pyr, NLC, and RXP-NLC. (**b**) Zeta potential distribution of NLC-S-S-Pyr, NLC, and RXP-NLC. (**c,d**) SEM images of NLC-S-S-Pyr after negative staining.

**Figure 3 molecules-24-03499-f003:**
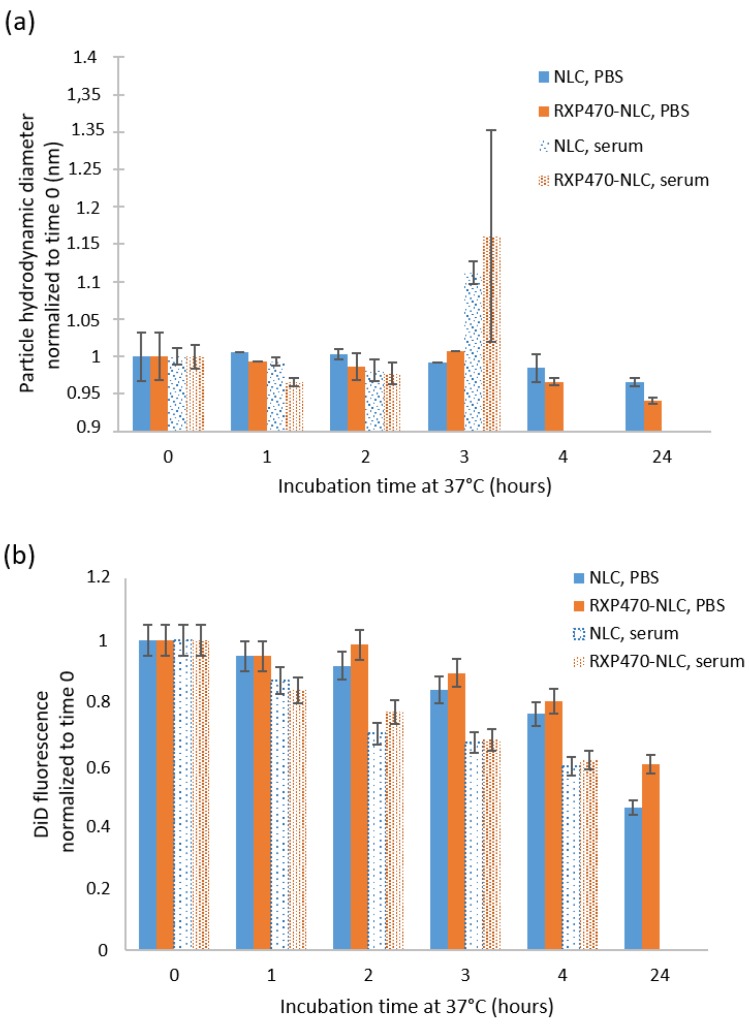
NLC and RXP-NLC stability when diluted at 5 mg/mL at 37 °C in 1X PBS buffer or in human serum. (**a**) Z-average particle diameter (nm) measured by DLS and normalized to value measured just after media and particle mixing. (**b**) DiD fluorescence (670 nm) normalized to value measured just after media and particle mixing.

**Figure 4 molecules-24-03499-f004:**
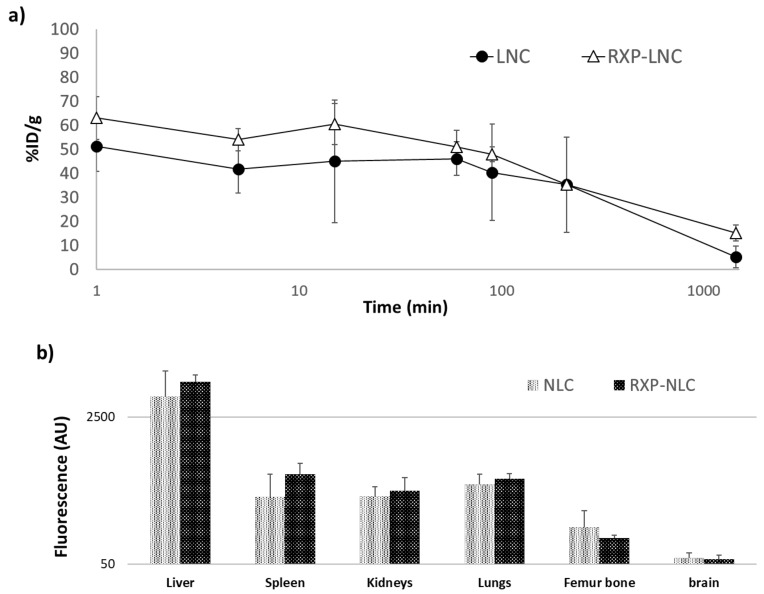
Blood clearance profiles and biodistribution for naked NLC and RXP-NLC in C57BL/6 ApoE^-/-^ mice (*n* = 3 for NLC, *n* = 3 for RXP-NLC 5 mg injected per mouse). (**a**) Comparison of blood clearance rates between the two NLCs (naked NLC: Black circles, RXP-NLC: Empty triangles). Time expressed in minutes is reported in logarithm scale. (**b**) Biodistribution and fluorescence intensity patterns observed for the two NLCs in liver, spleen, kidneys, lungs, femur bone, and brain harvested 24 h after injection. Fluorescence is expressed as arbitrary units (AU) and reported in logarithm scale.

**Figure 5 molecules-24-03499-f005:**
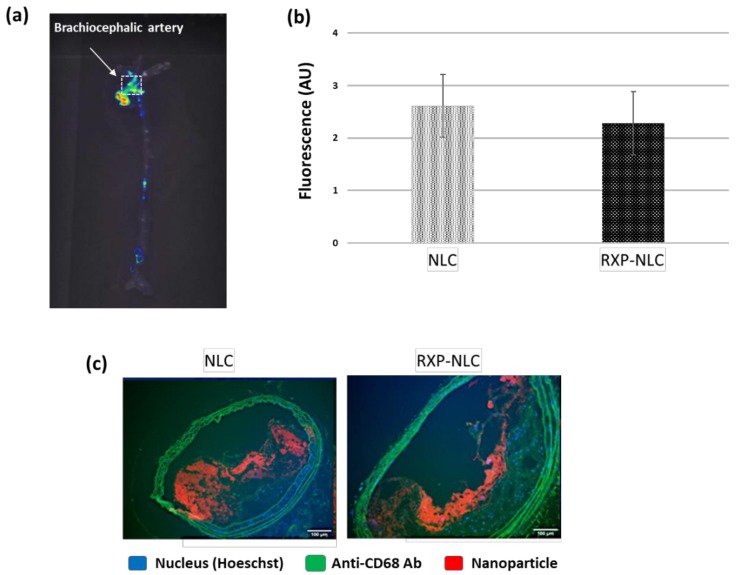
Tissue and histology analysis after NLC or RXP-NLC injection in ApoE^-/-^ mice. (**a**) Representative ex vivo fluorescence image of whole aorta harvested 24 h after intravenous injection of 5 mg particles/mouse. (**b**) Quantitative analysis of fluorescence signal within brachiocephalic artery (ROI = 1 mm^2^) and comparison between naked nanolipid carriers (NLC, *n* = 3) and RXP470.1 functionalized ones (RXP-NLC, *n* = 3). AU: Arbitrary units. (**c**) Aorta cross-section histology (fluorescence microscopy) of tissues harvested 48 h after intravenous injection. Blue: Nucleus (Hoeschst), Green: Anti-CD68 Ab (macrophage marker) and tissue auto-fluorescence, Red: DiD-loaded NLC. Scale bar: 100 µm.

**Table 1 molecules-24-03499-t001:** Characterization of NLC-S-S-Pyr, NLC, and RXP-NLC.

Nanoparticles	NLC-S-S-Pyr	NLC	RXP-NLC
Number of DiD/particle	-	70 ± 7	77 ± 8
Number of RXP/particle	-	-	105 ± 15
Hydrodynamic diameter (nm)	60.4 ± 1.3	62.1 ± 0.6	64.1 ± 1.5
Polydispersity index	0.15 ± 0.01	0.16 ± 0.02	0.19 ± 0.01
Zeta potential (mV)	−5.3 ± 7.5	−7.0 ± 4.3	−10.9 ± 6.4
Colloidal stability	-	>3 months ^1^	>3 months ^1^

^1^ Refer to Appendix A.

**Table 2 molecules-24-03499-t002:** Comparison of inhibition constants (Ki) between RXP470.1 and RXP-NLC towards a panel of eight human MMPs (hMMP-2, 3, 8, 9, 10, 12, 13, and 14). Ki (nM) values were determined in 50 mM Tris-HCl buffer, pH 6.8 with 10 mM CaCl_2_ at 25 °C. Ki values are the mean ± SD of three independent experiments. The selectivity factors were determined relative to hMMP12.

	hMMP-2	hMMP-3	hMMP-8	hMMP-9	hMMP-10	hMMP-12	hMMP-13	hMMP-14
**Ki (nM) RXP470.1**	52 ± 4	80 ± 2	120 ± 40	170 ± 50	17 ± 2	**0.26 ± 0.05**	11 ± 1	150 ± 21
**Selectivity factors**	200	308	462	654	65	**1**	42	577
**Ki (nM)** **RXP-NLC**	2892 ± 780	577 ± 140	>10,000	>10,000	136 ± 40	**2.7 ± 0.5**	295 ± 53	1053 ± 187
**Selectivity factors**	1071	213	>3700	>3700	50	**1**	109	390

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
