# Peer review of "Biodistribution of Nanostructured Lipid Carriers in Mice Atherosclerotic Model"

_molecules, 2019, doi:10.3390/molecules24193499_

Round 1

Reviewer 1 Report

Can you also show the characterisation data for NLC-S-S-Pyr starting material. diameter/pdi/zp. To confirm that no aggregation/changes occur during your functionalisation. Why is the selectivity factor 10-fold lower for RXP-NLC than RXP? Not discussed, and might provide an insight into why the RXP-NLC particles did not have any extra effect in vivo. much of discussion is redundant, some (such as new refs) can be moved to introduction if wished but largely just restates the intro. For example entire first paragraph, and also lines 280-289 where the authors discuss previous work before discussing own results. There is a significant contradiction in the discussion. Authors state:  The RXP470.1
292 pseudo peptide demonstrated its capacity to selectively block MMP12 proteolytic activity in several
293 preclinical models [42, 61-65]. Importantly, in ApoE-/- mice model of atherosclerosis, this compound
294 induced a change in atherosclerotic plaques composition thus limiting their evolution towards an
295 instable phenotype, strongly suggesting MMP12 as a relevant target in this model. In addition, optical
296 imaging agents with structure derived from that of RXP470.1 showed excellent in vivo performances
297 by targeting MMP12 within carotid arteries in a mouse model of aneurysm [66]

which discredits their later justification for lack of particle activity:

Like most of proteases,
312 MMP12 is activated only transiently leading to a very weak amount of proteolytically active forms.
313 Accordingly, this amount may be not sufficient to induce an active targeting that could result in an
314 improved uptake within atherosclerotic plaques.

This needs to be addressed. To me, it seems the problem lies with the particle (see point 2) rather than the receptor.

5. While the manuscript is generally well-written, there are a number of small spelling and grammatical errors. I would recommend that the manuscript is proof-read by a native English speaker to iron out these small mistakes before publication.

Author Response

Reviewer: 1

1) Can you also show the characterisation data for NLC-S-S-Pyr starting material. diameter/pdi/zp. To confirm that no aggregation/changes occur during your functionalization

REP: The data has been added in Table 1, and graphs displaying size distribution by intensity and zeta potential distribution have been added in Figure 2. All particle formulations appeared translucent, and no sign indicative of aggregation was observed throughout the experiments.

2) Why is the selectivity factor 10-fold lower for RXP-NLC than RXP? Not discussed, and might provide an insight into why the RXP-NLC particles did not have any extra effect in vivo

REP: As mentioned in the discussion part, the conception of RXP-NLC was directly inspired by previous results obtained with RXP470-derived optical probes validated in vivo (Razavian, Sci. Rep. 2016, Bordenave Bioconjugate Chemistry, 2016). In these studies, we indeed demonstrated that it was possible to install both a short PEG spacer and a fluorescent dye at the C-terminal end of RXP470 pseudo peptide without significantly impacting the affinity/selectivity profiles of resulting constructs. In vitro, RXP470-derived probes displayed a 10-fold lower affinity than the parent molecule for MMP-12 but their selectivity profile remained largely favorable for this enzyme. A comparable behavior is observed herein in the case of RXP-NLC with i) a drop in affinity towards MMP-12 of the same magnitude than previous optical probes, and ii) a selectivity profile preserved. RXP-NLC even display better selectivity factors than the parent RXP470 towards hMMP-2, 8, 9 and 13 (Table 2).

Both in the case of RXP470-derived probes and RXP-NLC, several parameters could affect the RXP470 binding efficiency towards MMP-12. A steric hindrance upon binding is not excluded but a PEG2 spacer was shown to be sufficient for limiting the impact of such a factor (see RX structure of a RXP470-derived probe in interaction with MMP-12 catalytic domain, Bordenave Bioconjugate Chemistry, 2016). In addition, the incorporation of a flexible PEG spacer could induce an additional entropic penalty upon ligand binding. In the specific case of RXP-NLC, the ligand density on the nanoparticle surface could not be optimal. For instance, a too high packing density could lead to a fraction of ligands inaccessible for binding to the targeted protein. 

Further optimizations both on the PEG spacer length and surface density of ligands could be then performed to restore a sub nanomolar affinity towards MMP-12. However, a gain in affinity in vitro does not ensure an improved uptake within the targeted tissues in vivo, since many other parameters could rule the in vivo performances of RXP-NLC (see also answer to point 4). The precise affinity at which maximum tissue uptake can be achieved remains difficult to predict and is often related both to the targeting molecule size and the nature of targeted tissues. In the case of small molecules, following their initial uptake, unbounded molecules are cleared rapidly from the tissues while bounded molecules are retained. Accordingly, a high level of specificity can be achieved when high affinity molecules are used. In contrast, several tumor targeting studies have shown that macromolecules in the size range of nanoparticles (radius >20 nm) could display similar tumor levels after injection of targeted and non-targeted macromolecules. In this case, the uptake is mainly governed by passive enhanced permeability and retention (EPR) effects with a slow clearance rate of unbound molecules by vascular intravasation comparable to that of antigen-bound molecules by cellular internalization (Schmidt Mol. Cancer Ther 2009). A similar phenomenon could occur when NLC accumulate within atherosclerotic plaques, preventing to visualize a significant impact of the targeting antigen.

Some elements of this answer have been included in the discussion part.

3) much of discussion is redundant, some (such as new refs) can be moved to introduction if wished but largely just restates the intro. For example entire first paragraph, and also lines 280-289 where the authors discuss previous work before discussing own results

REP: The introduction and discussion parts have been revised accordingly.

4) There is a significant contradiction in the discussion. Authors state: “The RXP470.1 pseudo peptide demonstrated its capacity to selectively block MMP12 proteolytic activity in several preclinical models [42, 61-65]. Importantly, in ApoE-/- mice model of atherosclerosis, this compound induced a change in atherosclerotic plaques composition thus limiting their evolution towards an instable phenotype, strongly suggesting MMP12 as a relevant target in

this model. In addition, optical imaging agents with structure derived from that of RXP470.1 showed excellent in vivo performances by targeting MMP12 within carotid arteries in a mouse model of aneurysm [66]” which discredits their later justification for lack of particle activity: “Like most of proteases, MMP12 is activated only transiently leading to a very weak amount of proteolytically active forms. Accordingly, this amount may be not sufficient to induce an active targeting that could result in an improved uptake within atherosclerotic plaques.”

 This needs to be addressed. To me, it seems the problem lies with the particle (see point 2) rather than the receptor

REP: Our study first demonstrates that RXP-NLC and naked NLC possess very similar in vivo performances. This already constitutes a valuable insight since the presence of RXP470 ligand could have increased the NLC opsonization with detrimental effect both on blood clearance profile and uptake within targeted tissues (Salvati Nat. Nanotechnol 2013, Xiao, International Journal of Pharmaceutic 2018). However, the targeting capability of the RXP-NLC does not appear superior to that of uncoated NLC.

In regards to the particles and as discussed above, both the size of the object and a non-optimal surface density of ligand may impair visualizing a significant targeting effect of RXP470. In addition, we have previously demonstrated that RXP470 ligand tends to bind serum albumin (Razavian, Sci. Rep. 2016, Bordenave Bioconjugate Chemistry, 2016). This may also occur with RXP-NLC leading to a fraction of targeting motifs potentially inaccessible for binding to MMP12.

On the other hand, the density of the MMP12 target may be also a limiting factor. Indeed, because uncontrolled Matrix Metalloprotease (MMP) activity could have disastrous effects on the microenvironment, MMPs activation is tightly controlled in vivo. Thus, over‐expression of MMPs reported in numerous pathological tissues mainly refers to MMPs under their inactive zymogen and inhibitor‐bound forms and only a small fraction of active form seems to be present (S. A. Sieber, Nat. Chem. Biol. 2006, 2, 274–281, D. Hesek, Chem Biol 2006). In this respect, we do not exclude that the amount of active MMP12 within atherosclerotic plaques, the sole form targeted by RXP470 ligand, may be not sufficient to induce a significant uptake of RXP-NLC relative to that of NLC.

This result does not seem contradictory with the results previously obtained with RXP470-derived optical probes and RXP470 inhibitor. As small peptides, these molecules indeed possess better diffusion properties within the tissues than RXP-NLC, with potentially positive impact on their targeting capacity.  

Importantly, the accumulation of NLC displaying RXP470 on their surface may allow inhibiting MMP12 activity, which could contribute to atherosclerotic plaques stabilization. This hypothesis is reinforced by the observation of RXP-NLC and uncoated NLC uptake in plaque area rich in macrophages that constitute the main source of MMP12 (Liang J, Circulation. 2006 Apr 25;113(16):1993-2001). A strategy that would consist in developing bifunctional NLC able to both achieve anti-inflammatory drug delivery and MMP12 blockade remains then relevant in the context of atherosclerotic plaque therapy.

A part of this answer has been included in the discussion part.

5) While the manuscript is generally well-written, there are a number of small spelling and grammatical errors. I would recommend that the manuscript is proof-read by a native English speaker to iron out these small mistakes before publication

REP : The manuscript has been carefully proof-read.

Reviewer: 2

      In their manuscript Devel et al., studied the biodistribution of nanostructured lipid carriers in mice atherosclerotic model. The paper is well written, and the purpose of the study is of great interest in the clinical setting for the optimal imaging of atherosclerotic lesions. The results are interesting. However, there are several major points linked to the essence of the work that require further clarification.

1) Introduction

2-3, lines 92-101 – not relevant. Why do you summarize your findings in this section? Please delete

REP: This paragraph was deleted and the introduction part revised accordingly.

line 55: paclitaxel…. and line 57: bromocriptine….– why do you use an ellipsis (three dots)? An ellipsis indicates an intentional omission of a word or phrase

REP: We replaced the three dots by e.g. before the examples

Moreover, is quercetin a drug for central nervous system diseases?

REP: Quercetin is an anti-oxidant and anti-inflammatory drug. It was loaded in solid lipid nanoparticles to improve its delivery to the brain with the objective to treat Alzheimer’s disease. These studies were performed in rats and reported by the group of Singh (Dhawan et al., J Pharm Pharmacol 63 342-351 (2011))

2, lines 73-74. “…could be accounted for by the….”. please correct

2, lines 75-78. I am sure if “since now” is correctly used in this sentence->

REP: These sentences were corrected, as well as other spelling and grammar errors.

2) Results/ Synthesis

Figure 1 is quite confusing for the reader. Although it depicts the synthesis of RXP470-modified NLC and control NLC the continuous route makes the reader believe only synthesis is achieved.

REP: Figure 2 was added in complement to Table 1 to display particle characterization.

How did you exactly calculate all the potential sites of binding for the ligand?

REP: Initially are introduced 2 µmol of SA-PEG100-S-S-Pyr for 750 mg of NLC-S-S-Pyr formulation. For each formulated NLC-S-S-Pyr batch, the particle concentration in the formulation is measured at the end of the fabrication/dialysis process (weighting after freeze-drying) to assess loss of product during the different manipulation (usually about 10-15 % loss, mainly during sonication (projection on the tube) and dialysis steps).

Based on the measurement of the particle concentration in the formulation, 300 mg of NLC-S-S-Pyr particles are then engaged in the coupling reaction, corresponding to 800 nmol of SA-PEG100-S-S-Pyr if the surfactant is lost in the same proportion than other lipid ingredients. In fact, we have demonstrated in another analytical study that weight loss during the preparation of NLC is mainly due to the elimination of the PEG fraction of Myrj s40 during dialysis (Myrj s40 is a mixture of PEG-stearate, PEG-palmitate, and PEG-OH without lipid chain for anchoring in the particle core) (Varache et al, Int J Pharm 2019, 566, 11-23).

Reactive thiol quantitation of NLC-SH obtained after deprotection of NLC-S-S-Pyr was also performed by fluorescence quantitation using excess of dye-maleimide conjugate (with appropriate controls to ensure that dye-maleimide coupling was specific to the presence of the SH function on the particle). In these experiments, more than 90% of theoretical SA-PEG100-S-S-Pyr reacted with dye-maleimide (more than 720 nmol of SA-PEG100-S-S-Pyr for 300 mg of particles).

Therefore, in the coupling reaction, the RXP ligand was introduced in a 1:1 ratio, with 720 nmol of RXP for 300 mg of particles, ie for 720 nmol of SA-PEG100-S-S-Pyr on the particle surface.

A DLS image is required to show the size distribution profile of the nanoparticles and to exclude the formation of aggregates (MAJOR)

REP: Data have been added in Table 1 for NLC-S-S-Pyr, and graphs displaying size distribution by intensity and zeta potential distribution have been added in Figure 2 a and b. All particle formulations appeared translucent, and no sign indicative of aggregation was observed throughout the experiments.

Did you perform microscopy (SEM) Please provide representative images (MAJOR).

REP: TEM images were added in Figure 2.

The zeta potential analysis of the nanoparticles should also be included, at least as supplementary

REP: The data have been added in Figure 2.

3) Biological application

Please change units (into hours) or scale (60 or 120 min) of x-axis in Figure 3a. You could also try a logarithmic scale in order to visual better the effect for the first 3 hours

REP: Figure 3a has been modified and a logarithm scale has been used to better visualize the NLC pharmacokinetics during the 3 first hours.

Figure 3a shows the detection of the NLC and RXP-NLC nanoparticles in the blood after 1 day. Did you also collected urines in order to see excretion?

REP: Urines were not collected during these experiments. However, all the published and unpublished previous biodistribution studies performed with similar NLC formulation in our group evidenced no signal in urines whatever time after IV injection. In particular refer to Mérian et al. J. Nucl. Med. 2013, 54, 1996-2003 for study in mice, and Sayag et al., Eur. J. Pharm. Biopharm. 2016, 100, 85-93 for study in dog.

Figure 3b. How does fluorescence intensity (AU) correlates with density of the nanoparticles? We can see that fluorescence is almost 20 times higher in the liver than in the spleen, kidneys and lungs and 100 times higher from the brain

As mentioned in the in vivo imaging experiments in mice (part 4.6), we first set up a calibration curve with DiD-loaded NLC. Different concentrations of NLC were incubated in blood samples and subsequently imaged by fluorescence reflectance using FMT1500’s planar imaging capability. From this calibration curve, blood fluorescence was converted to % injected dose per gram and allow establishing the nanoparticles PK. The density of nanoparticles accumulated within different organs could also be estimated. In this case, we would assume a minimum leakage of DiD from particles during circulation and bio distribution phases to correlate fluorescence intensity with nanoparticle density. Noteworthy, in the same animal model, we have previously shown that DiD alone was rapidly cleared from the blood within 30 min and displayed a renal clearance (unpublished data). This suggests a good in vivo stability for DiD-containing NLC that possess completely different in vivo properties. Further, the NLC biodistribution pattern is completely in line with previous studies: liver, adrenals and eventually ovaries (for female) are major organ of accumulation, far from other organs. In particular refer to Mérian et al. J. Nucl. Med. 2013, 54, 1996-2003 and Hirsjarvi and al. Nanomedicine: NBM 2013, 9, 375-387 (LNE particles) for study in mice, and Sayag et al., Eur. J. Pharm. Biopharm. 2016, 100, 85-93 for study in dog.

Figure 3b. Did you measure the fluorescence intensity of the nanoparticles in the adipose tissue

REP: Adipose tissue was not collected during these experiments. However, all the published and unpublished previous biodistribution studies performed in mice with similar NLC formulation in our group evidenced no signal in adipose tissue (noted as “fat”) whatever time after IV injection. In particular refer to Mérian et al. J. Nucl. Med. 2013, 54, 1996-2003 and Hirsjarvi and al. Nanomedicine: NBM 2013, 9, 375-387 (LNE particles)

4) Discussion

You coated the nanoparticles with a specific hMMP-12 inhibitor. However, you failed to notice any differences with the control NLC. Please elaborate more (MAJOR).

REP: Please refer to answer made to reviewer 1, points 2) and 4).

9, lines 274-275. If these nanoparticles like body lipid-rich areas (comment 9, results) then you should elaborate more on this and add more results if there are available (MAJOR)

REP:  In a previous study, we observed NLC accumulation in liver, as well as adrenals and ovaries (Mérian et al. J. Nucl. Med. 2013, 54, 1996-2003). Histology analysis of these organs revealed the nanoparticles were specifically located in the follicles and corpus luteum of the ovaries, and the X zone of adrenals, where steroid hormones are synthetized. They were also uptaken very efficiently in steroid-hormone-dependent tumors.

This could suggest that NLC once injected in blood could behave as chylomicrons/VLDL like particles acting as lipid nanotransporters. There are structural similarities between NLC and chylomicrons/VLDL (size and lipid composition: triglycerides in the core, phospholipids in the shell). Moreover, apolipoproteins have been demonstrated to be the main plasma proteins that interact with PEGylated lipid nanoparticles (Göppert et al, Eur J Pharm Biopharm 60, 361-372 (2005), and Int J Pharm 302, 172-186 (2005)). With apolipoprotein coating after their blood injection, NLC would become semi-synthetic mimic of chylomicrons/VLDL.

Disregulation of lipid metabolism is involved in atherosclerotic plaque formation and evolution. In particular, in early atherosclerotic plaque formation, LDL enter dysfunctional endothelium and accumulate in the arterial wall; this property is used to design contrast agents for plaque detection. HDL act as reverse-cholesterol transporter and present athero-protective properties. We could then expect that NLC, acting as semi-synthetic lipid transporters, mimic of chylomicrons/VLDL, could also interact with atherosclerotic plaques.

The previous sentence lines 274-275 was not clear. We modified it by “These observations tend to show this type of nanoparticles display after blood injection targeting properties towards body lipid-processing areas, as was suggested in a previous biodistribution study performed in healthy and cancer model mice [29].”

Is DiD dye toxic?

REP : DiD is non-toxic as such, but is not soluble at all in water. So solubilizing vehicules for DiD (DMSO, surfactant…) could be a source of toxicity. We have always observed an excellent biocompatibility of DiD when loaded in NLC: Mérian et al. J. Nucl. Med. 2013, 54, 1996-2003, Hirsjarvi and al. Nanomedicine: NBM 2013, and Gravier et al., J. Biomed. Opt. 2011, 16, 096013.

5) Methods

Please give the number of animal testing authorization issued by the relevant ethics committee.

REP: The authorization number for animal testing has been added

Why did you choose to perform the pharmacokinetics analysis and histology in two seperate experiments?

REP: Indeed, both pharmacokinetics and histology could have been performed on the same set of experiments. However, since histology analysis was carried out in a different location, and to avoid any sample damage during shipping, we decided to run these experiments independently. 

Reviewer 2 Report

In their manuscript Devel et al., studied the biodistribution of nanostructured lipid carriers in mice atherosclerotic model. The paper is well written, and the purpose of the study is of great interest in the clinical setting for the optimal imaging of atherosclerotic lesions. The results are interesting. However, there are several major points linked to the essence of the work that require further clarification.

Introduction

2-3, lines 92-101 – not relevant. Why do you summarize your findings in this section? Please delete. 2, line 55: paclitaxel…. and line 57: bromocriptine…. – why do you use an ellipsis (three dots)? An ellipsis indicates an intentional omission of a word or phrase. Moreover, is quercetin a drug for central nervous system diseases? 2, lines 73-74. “…could be accounted for by the….”. please correct 2, lines 75-78. I am sure if “since now” is correctly used in this sentence.

Results

Synthesis

Figure 1 is quite confusing for the reader. Although it depicts the synthesis of RXP470-modified NLC and control NLC the continuous route makes the reader believe only synthesis is achieved. How did you exactly calculate all the potential sites of binding for the ligand? A DLS image is required to show the size distribution profile of the nanoparticles and to exclude the formation of aggregates (MAJOR). Did you perform microscopy (SEM)? Please provide representative images (MAJOR). The zeta potential analysis of the nanoparticles should also be included, at least as supplementary.

Biological application

Please change units (into hours) or scale (60 or 120 min) of x-axis in Figure 3a. You could also try a logarithmic scale in order to visual better the effect for the first 3 hours. Figure 3a shows the detection of the NLC and RXP-NLC nanoparticles in the blood after 1 day. Did you also collected urines in order to see excretion? Figure 3b. How does fluorescence intensity (AU) correlates with density of the nanoparticles? We can see that fluorescence is almost 20 times higher in the liver than in the spleen, kidneys and lungs and 100 times higher from the brain. Figure 3b. Did you measure the fluorescence intensity of the nanoparticles in the adipose tissue?

Discussion

You coated the nanoparticles with a specific hMMP-12 inhibitor. However, you failed to notice any differences with the control NLC. Please elaborate more (MAJOR). 9, lines 274-275. If these nanoparticles like body lipid-rich areas (comment 9, results) then you should elaborate more on this and add more results if there are available (MAJOR). Is DiD dye toxic?

Methods

Please give the number of animal testing authorization issued by the relevant ethics committee. Why did you choose to perform the pharmacokinetics analysis and histology in two seperate experiments?

Author Response

Reviewer: 2

      In their manuscript Devel et al., studied the biodistribution of nanostructured lipid carriers in mice atherosclerotic model. The paper is well written, and the purpose of the study is of great interest in the clinical setting for the optimal imaging of atherosclerotic lesions. The results are interesting. However, there are several major points linked to the essence of the work that require further clarification.

1) Introduction

2-3, lines 92-101 – not relevant. Why do you summarize your findings in this section? Please delete

REP: This paragraph was deleted and the introduction part revised accordingly.

line 55: paclitaxel…. and line 57: bromocriptine….– why do you use an ellipsis (three dots)? An ellipsis indicates an intentional omission of a word or phrase

REP: We replaced the three dots by e.g. before the examples

Moreover, is quercetin a drug for central nervous system diseases?

REP: Quercetin is an anti-oxidant and anti-inflammatory drug. It was loaded in solid lipid nanoparticles to improve its delivery to the brain with the objective to treat Alzheimer’s disease. These studies were performed in rats and reported by the group of Singh (Dhawan et al., J Pharm Pharmacol 63 342-351 (2011))

2, lines 73-74. “…could be accounted for by the….”. please correct

2, lines 75-78. I am sure if “since now” is correctly used in this sentence->

REP: These sentences were corrected, as well as other spelling and grammar errors.

2) Results/ Synthesis

Figure 1 is quite confusing for the reader. Although it depicts the synthesis of RXP470-modified NLC and control NLC the continuous route makes the reader believe only synthesis is achieved.

REP: Figure 2 was added in complement to Table 1 to display particle characterization.

How did you exactly calculate all the potential sites of binding for the ligand?

REP: Initially are introduced 2 µmol of SA-PEG100-S-S-Pyr for 750 mg of NLC-S-S-Pyr formulation. For each formulated NLC-S-S-Pyr batch, the particle concentration in the formulation is measured at the end of the fabrication/dialysis process (weighting after freeze-drying) to assess loss of product during the different manipulation (usually about 10-15 % loss, mainly during sonication (projection on the tube) and dialysis steps).

Based on the measurement of the particle concentration in the formulation, 300 mg of NLC-S-S-Pyr particles are then engaged in the coupling reaction, corresponding to 800 nmol of SA-PEG100-S-S-Pyr if the surfactant is lost in the same proportion than other lipid ingredients. In fact, we have demonstrated in another analytical study that weight loss during the preparation of NLC is mainly due to the elimination of the PEG fraction of Myrj s40 during dialysis (Myrj s40 is a mixture of PEG-stearate, PEG-palmitate, and PEG-OH without lipid chain for anchoring in the particle core) (Varache et al, Int J Pharm 2019, 566, 11-23).

Reactive thiol quantitation of NLC-SH obtained after deprotection of NLC-S-S-Pyr was also performed by fluorescence quantitation using excess of dye-maleimide conjugate (with appropriate controls to ensure that dye-maleimide coupling was specific to the presence of the SH function on the particle). In these experiments, more than 90% of theoretical SA-PEG100-S-S-Pyr reacted with dye-maleimide (more than 720 nmol of SA-PEG100-S-S-Pyr for 300 mg of particles).

Therefore, in the coupling reaction, the RXP ligand was introduced in a 1:1 ratio, with 720 nmol of RXP for 300 mg of particles, ie for 720 nmol of SA-PEG100-S-S-Pyr on the particle surface.

A DLS image is required to show the size distribution profile of the nanoparticles and to exclude the formation of aggregates (MAJOR)

REP: Data have been added in Table 1 for NLC-S-S-Pyr, and graphs displaying size distribution by intensity and zeta potential distribution have been added in Figure 2 a and b. All particle formulations appeared translucent, and no sign indicative of aggregation was observed throughout the experiments.

Did you perform microscopy (SEM) Please provide representative images (MAJOR).

REP: TEM images were added in Figure 2.

The zeta potential analysis of the nanoparticles should also be included, at least as supplementary

REP: The data have been added in Figure 2.

3) Biological application

Please change units (into hours) or scale (60 or 120 min) of x-axis in Figure 3a. You could also try a logarithmic scale in order to visual better the effect for the first 3 hours

REP: Figure 3a has been modified and a logarithm scale has been used to better visualize the NLC pharmacokinetics during the 3 first hours.

Figure 3a shows the detection of the NLC and RXP-NLC nanoparticles in the blood after 1 day. Did you also collected urines in order to see excretion?

REP: Urines were not collected during these experiments. However, all the published and unpublished previous biodistribution studies performed with similar NLC formulation in our group evidenced no signal in urines whatever time after IV injection. In particular refer to Mérian et al. J. Nucl. Med. 2013, 54, 1996-2003 for study in mice, and Sayag et al., Eur. J. Pharm. Biopharm. 2016, 100, 85-93 for study in dog.

Figure 3b. How does fluorescence intensity (AU) correlates with density of the nanoparticles? We can see that fluorescence is almost 20 times higher in the liver than in the spleen, kidneys and lungs and 100 times higher from the brain

As mentioned in the in vivo imaging experiments in mice (part 4.6), we first set up a calibration curve with DiD-loaded NLC. Different concentrations of NLC were incubated in blood samples and subsequently imaged by fluorescence reflectance using FMT1500’s planar imaging capability. From this calibration curve, blood fluorescence was converted to % injected dose per gram and allow establishing the nanoparticles PK. The density of nanoparticles accumulated within different organs could also be estimated. In this case, we would assume a minimum leakage of DiD from particles during circulation and bio distribution phases to correlate fluorescence intensity with nanoparticle density. Noteworthy, in the same animal model, we have previously shown that DiD alone was rapidly cleared from the blood within 30 min and displayed a renal clearance (unpublished data). This suggests a good in vivo stability for DiD-containing NLC that possess completely different in vivo properties. Further, the NLC biodistribution pattern is completely in line with previous studies: liver, adrenals and eventually ovaries (for female) are major organ of accumulation, far from other organs. In particular refer to Mérian et al. J. Nucl. Med. 2013, 54, 1996-2003 and Hirsjarvi and al. Nanomedicine: NBM 2013, 9, 375-387 (LNE particles) for study in mice, and Sayag et al., Eur. J. Pharm. Biopharm. 2016, 100, 85-93 for study in dog.

Figure 3b. Did you measure the fluorescence intensity of the nanoparticles in the adipose tissue

REP: Adipose tissue was not collected during these experiments. However, all the published and unpublished previous biodistribution studies performed in mice with similar NLC formulation in our group evidenced no signal in adipose tissue (noted as “fat”) whatever time after IV injection. In particular refer to Mérian et al. J. Nucl. Med. 2013, 54, 1996-2003 and Hirsjarvi and al. Nanomedicine: NBM 2013, 9, 375-387 (LNE particles)

4) Discussion

You coated the nanoparticles with a specific hMMP-12 inhibitor. However, you failed to notice any differences with the control NLC. Please elaborate more (MAJOR).

REP: Please refer to answer made to reviewer 1, points 2) and 4) (see below).

9, lines 274-275. If these nanoparticles like body lipid-rich areas (comment 9, results) then you should elaborate more on this and add more results if there are available (MAJOR)

REP:  In a previous study, we observed NLC accumulation in liver, as well as adrenals and ovaries (Mérian et al. J. Nucl. Med. 2013, 54, 1996-2003). Histology analysis of these organs revealed the nanoparticles were specifically located in the follicles and corpus luteum of the ovaries, and the X zone of adrenals, where steroid hormones are synthetized. They were also uptaken very efficiently in steroid-hormone-dependent tumors.

This could suggest that NLC once injected in blood could behave as chylomicrons/VLDL like particles acting as lipid nanotransporters. There are structural similarities between NLC and chylomicrons/VLDL (size and lipid composition: triglycerides in the core, phospholipids in the shell). Moreover, apolipoproteins have been demonstrated to be the main plasma proteins that interact with PEGylated lipid nanoparticles (Göppert et al, Eur J Pharm Biopharm 60, 361-372 (2005), and Int J Pharm 302, 172-186 (2005)). With apolipoprotein coating after their blood injection, NLC would become semi-synthetic mimic of chylomicrons/VLDL.

Disregulation of lipid metabolism is involved in atherosclerotic plaque formation and evolution. In particular, in early atherosclerotic plaque formation, LDL enter dysfunctional endothelium and accumulate in the arterial wall; this property is used to design contrast agents for plaque detection. HDL act as reverse-cholesterol transporter and present athero-protective properties. We could then expect that NLC, acting as semi-synthetic lipid transporters, mimic of chylomicrons/VLDL, could also interact with atherosclerotic plaques.

The previous sentence lines 274-275 was not clear. We modified it by “These observations tend to show this type of nanoparticles display after blood injection targeting properties towards body lipid-processing areas, as was suggested in a previous biodistribution study performed in healthy and cancer model mice [29].”

Is DiD dye toxic?

REP : DiD is non-toxic as such, but is not soluble at all in water. So solubilizing vehicules for DiD (DMSO, surfactant…) could be a source of toxicity. We have always observed an excellent biocompatibility of DiD when loaded in NLC: Mérian et al. J. Nucl. Med. 2013, 54, 1996-2003, Hirsjarvi and al. Nanomedicine: NBM 2013, and Gravier et al., J. Biomed. Opt. 2011, 16, 096013.

5) Methods

Please give the number of animal testing authorization issued by the relevant ethics committee.

REP: The authorization number for animal testing has been added

Why did you choose to perform the pharmacokinetics analysis and histology in two seperate experiments?

REP: Indeed, both pharmacokinetics and histology could have been performed on the same set of experiments. However, since histology analysis was carried out in a different location, and to avoid any sample damage during shipping, we decided to run these experiments independently. 

 Reviewer: 1

1) Can you also show the characterisation data for NLC-S-S-Pyr starting material. diameter/pdi/zp. To confirm that no aggregation/changes occur during your functionalization

REP: The data has been added in Table 1, and graphs displaying size distribution by intensity and zeta potential distribution have been added in Figure 2. All particle formulations appeared translucent, and no sign indicative of aggregation was observed throughout the experiments.

2) Why is the selectivity factor 10-fold lower for RXP-NLC than RXP? Not discussed, and might provide an insight into why the RXP-NLC particles did not have any extra effect in vivo

REP: As mentioned in the discussion part, the conception of RXP-NLC was directly inspired by previous results obtained with RXP470-derived optical probes validated in vivo (Razavian, Sci. Rep. 2016, Bordenave Bioconjugate Chemistry, 2016). In these studies, we indeed demonstrated that it was possible to install both a short PEG spacer and a fluorescent dye at the C-terminal end of RXP470 pseudo peptide without significantly impacting the affinity/selectivity profiles of resulting constructs. In vitro, RXP470-derived probes displayed a 10-fold lower affinity than the parent molecule for MMP-12 but their selectivity profile remained largely favorable for this enzyme. A comparable behavior is observed herein in the case of RXP-NLC with i) a drop in affinity towards MMP-12 of the same magnitude than previous optical probes, and ii) a selectivity profile preserved. RXP-NLC even display better selectivity factors than the parent RXP470 towards hMMP-2, 8, 9 and 13 (Table 2).

Both in the case of RXP470-derived probes and RXP-NLC, several parameters could affect the RXP470 binding efficiency towards MMP-12. A steric hindrance upon binding is not excluded but a PEG2 spacer was shown to be sufficient for limiting the impact of such a factor (see RX structure of a RXP470-derived probe in interaction with MMP-12 catalytic domain, Bordenave Bioconjugate Chemistry, 2016). In addition, the incorporation of a flexible PEG spacer could induce an additional entropic penalty upon ligand binding. In the specific case of RXP-NLC, the ligand density on the nanoparticle surface could not be optimal. For instance, a too high packing density could lead to a fraction of ligands inaccessible for binding to the targeted protein. 

Further optimizations both on the PEG spacer length and surface density of ligands could be then performed to restore a sub nanomolar affinity towards MMP-12. However, a gain in affinity in vitro does not ensure an improved uptake within the targeted tissues in vivo, since many other parameters could rule the in vivo performances of RXP-NLC (see also answer to point 4). The precise affinity at which maximum tissue uptake can be achieved remains difficult to predict and is often related both to the targeting molecule size and the nature of targeted tissues. In the case of small molecules, following their initial uptake, unbounded molecules are cleared rapidly from the tissues while bounded molecules are retained. Accordingly, a high level of specificity can be achieved when high affinity molecules are used. In contrast, several tumor targeting studies have shown that macromolecules in the size range of nanoparticles (radius >20 nm) could display similar tumor levels after injection of targeted and non-targeted macromolecules. In this case, the uptake is mainly governed by passive enhanced permeability and retention (EPR) effects with a slow clearance rate of unbound molecules by vascular intravasation comparable to that of antigen-bound molecules by cellular internalization (Schmidt Mol. Cancer Ther 2009). A similar phenomenon could occur when NLC accumulate within atherosclerotic plaques, preventing to visualize a significant impact of the targeting antigen.

Some elements of this answer have been included in the discussion part.

3) much of discussion is redundant, some (such as new refs) can be moved to introduction if wished but largely just restates the intro. For example entire first paragraph, and also lines 280-289 where the authors discuss previous work before discussing own results

REP: The introduction and discussion parts have been revised accordingly.

4) There is a significant contradiction in the discussion. Authors state: “The RXP470.1 pseudo peptide demonstrated its capacity to selectively block MMP12 proteolytic activity in several preclinical models [42, 61-65]. Importantly, in ApoE-/- mice model of atherosclerosis, this compound induced a change in atherosclerotic plaques composition thus limiting their evolution towards an instable phenotype, strongly suggesting MMP12 as a relevant target in

this model. In addition, optical imaging agents with structure derived from that of RXP470.1 showed excellent in vivo performances by targeting MMP12 within carotid arteries in a mouse model of aneurysm [66]” which discredits their later justification for lack of particle activity: “Like most of proteases, MMP12 is activated only transiently leading to a very weak amount of proteolytically active forms. Accordingly, this amount may be not sufficient to induce an active targeting that could result in an improved uptake within atherosclerotic plaques.”

 This needs to be addressed. To me, it seems the problem lies with the particle (see point 2) rather than the receptor

REP: Our study first demonstrates that RXP-NLC and naked NLC possess very similar in vivo performances. This already constitutes a valuable insight since the presence of RXP470 ligand could have increased the NLC opsonization with detrimental effect both on blood clearance profile and uptake within targeted tissues (Salvati Nat. Nanotechnol 2013, Xiao, International Journal of Pharmaceutic 2018). However, the targeting capability of the RXP-NLC does not appear superior to that of uncoated NLC.

In regards to the particles and as discussed above, both the size of the object and a non-optimal surface density of ligand may impair visualizing a significant targeting effect of RXP470. In addition, we have previously demonstrated that RXP470 ligand tends to bind serum albumin (Razavian, Sci. Rep. 2016, Bordenave Bioconjugate Chemistry, 2016). This may also occur with RXP-NLC leading to a fraction of targeting motifs potentially inaccessible for binding to MMP12.

On the other hand, the density of the MMP12 target may be also a limiting factor. Indeed, because uncontrolled Matrix Metalloprotease (MMP) activity could have disastrous effects on the microenvironment, MMPs activation is tightly controlled in vivo. Thus, over‐expression of MMPs reported in numerous pathological tissues mainly refers to MMPs under their inactive zymogen and inhibitor‐bound forms and only a small fraction of active form seems to be present (S. A. Sieber, Nat. Chem. Biol. 2006, 2, 274–281, D. Hesek, Chem Biol 2006). In this respect, we do not exclude that the amount of active MMP12 within atherosclerotic plaques, the sole form targeted by RXP470 ligand, may be not sufficient to induce a significant uptake of RXP-NLC relative to that of NLC.

This result does not seem contradictory with the results previously obtained with RXP470-derived optical probes and RXP470 inhibitor. As small peptides, these molecules indeed possess better diffusion properties within the tissues than RXP-NLC, with potentially positive impact on their targeting capacity.  

Importantly, the accumulation of NLC displaying RXP470 on their surface may allow inhibiting MMP12 activity, which could contribute to atherosclerotic plaques stabilization. This hypothesis is reinforced by the observation of RXP-NLC and uncoated NLC uptake in plaque area rich in macrophages that constitute the main source of MMP12 (Liang J, Circulation. 2006 Apr 25;113(16):1993-2001). A strategy that would consist in developing bifunctional NLC able to both achieve anti-inflammatory drug delivery and MMP12 blockade remains then relevant in the context of atherosclerotic plaque therapy.

A part of this answer has been included in the discussion part.

5) While the manuscript is generally well-written, there are a number of small spelling and grammatical errors. I would recommend that the manuscript is proof-read by a native English speaker to iron out these small mistakes before publication

REP : The manuscript has been carefully proof-read.
